# From fragmentation to resilience: Connectivity and habitat diversity as drivers of fish persistence in California watersheds

Jeanette K. Howard[1]*, Analie R. Barnett[2]☯, Kurt A. Fesenmyer[3]☯, Mark G. Anderson[4]‡

**1** The Nature Conservancy, San Francisco, California, United States of America, **2** Center for Resilient Conservation Science, The Nature Conservancy, Atlanta, Georgia, United States of America, **3** The Nature Conservancy, Boise, Idaho, United States of America, **4** Center for Resilient Conservation Science, The Nature Conservancy, Newburyport, Massachusetts, United States of America

☯ These authors contributed equally to this work
‡ This author also contributed equally to this work.
* jeanette.howard@gmail.com

## Abstract

This study evaluates how well key elements of freshwater resilience (e.g., hydrographic, physical habitat, and condition variables) explain the persistence of native fish species over time. Using the Temporal Beta Index (TBI), we quantify the change in fish species presence-absence in functionally connected networks within California to determine which watersheds within the network experienced significant changes in fish community composition. Random forest (RF) models were used to explore how the suite of network attributes influenced TBI and how the relationships varied by ecoregion. By integrating historical and contemporary fish distribution records with comprehensive datasets on fish passage barriers, stream habitat typologies, and watershed conditions, the analysis provides evidence that fragmentation—primarily driven by a century of dam construction—has impacted the persistence of fish species throughout the state. These results underscore the importance of maintaining and restoring interconnected river networks to preserve habitat heterogeneity, ensure the continued functionality of freshwater processes, and promote long-term ecological stability amidst ongoing and future environmental challenges. This research provides a framework to evaluate what factors contributed to fish loss in the past, thereby offering insights into enhancing the resilience of freshwater ecosystems and persistence of freshwater species into the future.

## Introduction

Maintaining connected landscapes is the most widely cited strategy for building climate change resilience [1]. This may be especially true for river networks where fragmentation disrupts the interconnected physical, chemical and biological building

**Data availability statement:** Spatial and tabular data for reproducing this analysis are available at doi.org/10.6084/m9.figshare.30096151.

**Funding:** The author(s) received no specific funding for this work.

**Competing interests:** The authors have declared that no competing interests exist.

blocks of river ecosystems. Connectivity enables processes such as flow, sediment, and nutrient regimes to function naturally and is crucial in determining habitat availability, composition and distribution of aquatic communities [2,3]. Longer, more connected stream networks contain a diversity of physical habitat conditions and together with water availability, drive resilience in freshwater systems by providing habitat heterogeneity and the conditions to maintain ecological function [4]. Connectivity allows species to move throughout the network to find feeding and spawning habitat, and, in times of stress, relocate to more suitable conditions creating resilience [5–8].

Freshwater species are uniquely vulnerable to changes in climate and habitat degradation because they are confined to aquatic habitats where movement to alternative habitats is typically more restricted than in terrestrial systems [9]. Longer connected riverine networks are predicted to be more resilient to climate change as connectivity helps regulate riverine processes that maintain function and diversity (such as flow, temperature, water quality and food webs). Longitudinal connectivity determines habitat type and availability for river-dependent taxa most notably fish species that utilize dendritic networks throughout their life cycles [10]. Dams and other barriers are the primary source of the loss of connectivity and increase in fragmentation, which alters ecosystem processes, functions and biotic communities [6,11–13]. Dams and other barriers can lead to isolation of fish populations [14], reduce potential for recolonization and metapopulation persistence [3,15], change species composition [16,17] and even result in extirpation [18,19].

North America's stream networks and associated freshwater biota are expected to change substantially over the next century in concert with changes in precipitation and temperature [20,21]. Assessing and mapping characteristics of freshwater systems that have conveyed resilience in the past may help determine site resilience into the future. Resilience is the ability of these systems to absorb and recover from changes and to maintain diversity and function even as its composition changes in response to climate change [22]. Core to resilience is the capacity of ecosystems to adapt to a changing climate while maintaining function and diversity [23].

The well-studied relationship between habitat diversity and species biodiversity within freshwater systems [24–26] provides a strong conceptual basis for conservation strategies that seek to ensure freshwater biodiversity persistence by protecting remaining, large, connected networks and restoring connectivity to fragmented networks. However, studies evaluating if connectivity and habitat diversity affect fish community composition and individual species persistence at the landscape scale are lacking. In this study, we examine if changes in fish communities and individual species persistence are linked to changes in network connectivity and habitat diversity in California watersheds. We use a unique dataset of contemporary and historical fish species distribution in California [27], comprehensive fish passage barrier datasets [28,29], stream habitat typing datasets [30], and watershed condition datasets [31] to quantify the factors that are shown to be the most important to fish persistence in California's river networks stratified by freshwater ecoregions [32].

In California, native freshwater taxa struggle to persist due to a 100 + year history of extensive dam construction, surface water diversion, natural land cover conversion, and ground water overdraft, which have increased fragmentation and loss of connectivity and decreased watershed condition. As a result of these changes, along with deteriorating water quality and introduced aquatic species, over half of California's freshwater taxa are considered vulnerable to extinction [33] and 83% of California's native freshwater fishes are considered at risk of extinction in the next century [34].

This study evaluates how well hydrographic, physical habitat, and condition variables - key elements of freshwater resilience [4] - explain the persistence of native fish species over time. Specifically, we quantify the change in fish species presence-absence in watersheds within four North American freshwater ecoregions to determine which watersheds experienced significant changes in fish community composition over time, whether those changes were dominated by species losses or gains, and which species experienced significant changes. Using datasets of stream-river network attributes that characterize change in physical habitat availability over time as well as the flow alteration and watershed condition of current networks, we identify the potential drivers of fish community change and species loss over time and discuss the implications for future freshwater conservation and restoration work.

We hypothesize that longer stream networks that contain increased habitat heterogeneity (e.g., size class variation, temperature gradients, substrate variety, etc.) can better accommodate change while maintaining a functional ecological state, and thus support biodiversity and in this case, fish species persistence in California.

## Methods

### Study area

Our study area covers 471,500 km$^2$ of California and adjacent, hydrologically connected portions of Oregon, Nevada, and Arizona in the USA. Primary river basins included in the study area are the Klamath, Sacramento, San Joaquin, Amargosa, Truckee, and Colorado River systems. We initially delineated six ecoregions within the study area for stratification purposes based on hydrological boundaries and the Freshwater Ecoregions of the World [32]. After reviewing the available fish species data and considering their size and similar characteristics, we combined the three arid ecoregions into a single ecoregion. Final ecoregions include Northern California, Sacramento-San Joaquin, Deserts-Lahontan and Southern California (Fig 1).

### Hydrography, Physical Habitat, and Condition Attributes

We used 1:100,000 scale National Hydrography Dataset (NHD) Plus v2 stream reaches [35] as the base hydrography to which attributes were assigned. The basic unit of the dataset is the stream reach, which represents a confluence-to-confluence segment of a river network. NHDPlus v2 and other derived products provide attributes of local characteristics for each stream reach and characteristics for the contributing upstream area for each stream reach. We classified each stream reach into physical habitat types related to its local characteristics (gradient, presence of water body, and valley confinement) and contributing upstream area characteristics (drainage area, temperature, and flow regime). We relied on NHDPlus v2 for a continuous measure of drainage area and reclassified those values into stream size classes described by McManamay and others [30]. We used a conterminous US stream classification [36] developed for NHDPlus v2 reaches to categorize gradient, valley confinement, and flow regime (S1 Table). We used a continuous, regional prediction of average August stream temperature [37] to represent thermal regime in each reach and classified these values using categories from the conterminous US stream classification system [36]. Regional stream temperature predictions were available in all freshwater ecoregions except portions of the Deserts-Lahontan ecoregion; in the portion of that ecoregion lacking regional stream temperature predictions, we used the conterminous US stream classification of July-August stream temperature [36]. Additionally, we created reach classes for indicating presence of three water body types: ocean/estuary, natural lakes greater than 2.5 km$^2$, and artificial lakes greater than 2.5 km$^2$. Stream reaches with drainage areas < 2.5 km$^2$ were excluded from the analysis to minimize variable drainage densities within the NHDPlus v2.

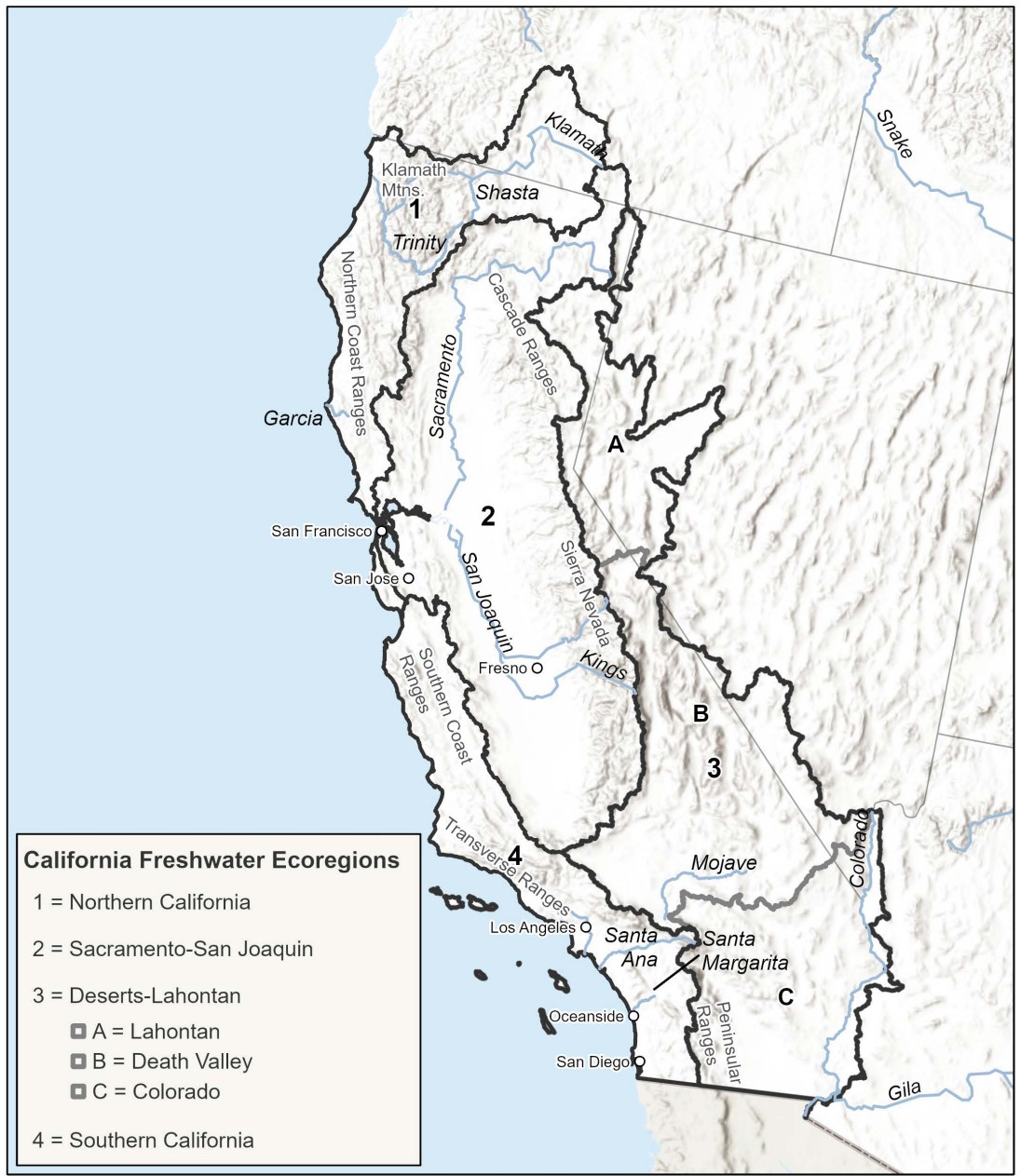

**Fig 1. Map of Study Area.** California freshwater ecoregions used in the analysis are depicted with a black outline. Gray boundaries show the three arid ecoregions of Lahontan, Death Valley, and Colorado that were combined into a single ecoregion named Deserts-Lahontan. Rivers discussed in the paper are shown for reference as are major cities and mountain ranges. World Hillshade is used in the figure's background (Sources: Esri, Airbus DS, USGS, NGA, NASA, CGIAR, N Robinson, NCEAS, NLS, OS, NMA, Geodatastyrelsen, Rijkswaterstaat, GSA, Geoland, FEMA, Intermap, and the GIS User Community). The functionally connected networks were generated using the National Hydrography Dataset (NHD) Plus v2 stream reaches (https://www.epa.gov/waterdata/nhdplus-national-hydrography-dataset-plus) [35]; the waterfalls data was sourced from the National Hydrography Dataset (NHD) Plus v2 (https://www.epa.gov/waterdata/nhdplus-national-hydrography-dataset-plus) [35]; the dams data was sourced from the National Anthropogenic Barrier Dataset (https://www.sciencebase.gov/catalog/item/56a7f9dce4b0b28f1184dabd) [43]; subwatershed boundaries were generated from USDA-USGS-EPA Watershed Boundary Dataset, a companion dataset of the National Hydrography Dataset [35]. All sources are in the public domain. These maps were created by using ArcGIS Pro 3.5.2 (ESRI, Redlands, CA).

We relied on two datasets for characterizing stream habitat condition for each stream reach, California's Integrated Assessment of Watershed Health [38] and USEPA's StreamCat [39]. California's Integrated Assessment of Watershed Health synthesized information on landscape characteristics (e.g., % natural land cover, road crossing density, sedimentation risk) to delineate an index of habitat condition [38]. The Watershed Condition Index, hereafter "HWI" is calculated as the average of rank-normalized metrics with higher index scores corresponding to higher instream habitat condition. To characterize habitat impairment associated with water infrastructure development, we used StreamCat's national hydrologic regulation index [40], a measure of the presence and volume of reservoir, agriculture and impervious land cover, and the density of canals and ditches [39] at the scale of individual stream reaches ("local" scale). We also used StreamCat measures of dam density and dam storage for the total upstream contributing area of each stream reach ("upstream" scale).

## Passage Barriers

To assess the impact of dam construction on historic connectivity, we compiled state and national datasets of potential or known natural and artificial barriers to aquatic organism passage from several sources. For natural barriers, we identified waterfalls characterized as complete barriers to fish passage in California's Passage Assessment Database [28], Oregon Fish Passage Barrier dataset [41], and the Lahontan cutthroat trout barrier dataset [42]. We supplemented these data with waterfalls in NHDPlus v2 [35]. For artificial barriers, we used dams identified as complete, partial, or unknown barriers to fish passage in a national dataset ([43] and the three regional datasets noted above. We excluded dams that did not fall along the routed portion of the NHDPlus v2 streams (e.g., the dam was off-channel water storage) and dams identified as removed by a national database [44]. For both natural and artificial barriers in the Northern California, Sacramento-San Joaquin, and Southern California ecoregions, we further excluded waterfalls or dams with observed anadromous fish distribution ([45,46] upstream of the barrier. We did not include road culverts or surface water diversions due to incomplete coverage of inventory data for these features in the study area and the impermanence of these features relative to dams and waterfalls.

## Functionally Connected Networks

We used the passage barriers dataset and the NHDPlus v2 data structure to split the river network into connected habitat patches by navigating downstream from headwaters until reaching a passage barrier or network terminus (e.g., ocean or hydrological disconnect in NHDPlus v2), aggregating the upstream reaches (inclusive of reaches with barrier or terminus), and iteratively moving to the next unaggregated reach using the methods developed by Nathan et al. (2020) [47]. We refer to the habitat patches generated by this process as functionally connected networks (FCNs) and created versions representing historical conditions, using waterfalls only, and current conditions, using waterfalls and dams.

After delineating historical and current FCNs, we summed the total length of each physical habitat class for each FCN. We considered any physical habitat class to be present within an FCN if the total length of the class was at least 1 km. Water bodies (ocean/estuaries and natural or artificial lakes) did not have a minimum length requirement and only needed to be present. For each FCN, we summed the total number of habitat classes present to calculate physical habitat richness and we calculated rarity-weighted habitat richness as the sum of the inverse of the number of FCNs that contain each class of physical habitat diversity [48]. In the calculation of the rarity-weighted habitat richness value, the denominator is the number of networks with an individual habitat type. The number of networks in each ecoregion is different for the two time periods, with more networks present during the current time period due to fragmentation from dams. Thus, an increase in the number of networks results in smaller rarity-weighted richness values, making it difficult to compare the historic and current rare habitat richness values. To address this issue, we translated the rarity-weighted habitat richness values to z-scores by ecoregion (S6 Table) and calculated the change in z-scores from current to historic.

## Historical and Current Fish Distribution

We used the geospatial database and mapping software system PISCES (http://pisces.ucdavis.edu/) to estimate the historical and current range of freshwater fish taxa in California [27]. The PISCES fish data compiled expert informed range maps and ~300,000 records of native fish occurrences in primary and gray literature to derive best estimates of historical and current ranges of California's 129 native freshwater fish taxa at the 12-digit hydrologic unit code (HUC-12, or subwatershed) scale (S2 Table). Observations of fish prior to 1975 were used to estimate historical taxa ranges, while those after 1975 represent current ranges. We did not consider translocations within the current range of species. For each species, we also referenced a previously published categorization of range type as anadromous, narrow-ranging (<24,000 km$^2$ total range), or wide-ranging (>24,000 km$^2$) [49].

To determine if any HUC-12s experienced a significant change in native fish species composition from historic to current periods within each ecoregion, we calculated a temporal beta-diversity index (TBI) [50] using the *adespatial* R package [51] with the PISCES fish presence-absence data, Sorensen dissimilarity coefficient, and 9,999 permutations. All statistical analyses were conducted by freshwater ecoregion (Fig 1) and were performed using R Statistical Software (v4.1.3; R Core Team 2022). Low TBI values indicate lower change in fish species composition, while higher TBI values indicate greater change. The null hypothesis is that a HUC-12 is not exceptionally different between two time periods for presence-absence data, compared to randomly generated assemblages that could have been observed in the HUC-12 for the two time periods. We also used the fish presence-absence data in a paired-t test permutation using the *tpaired. krandtest R* function in *adespatial*, with a correction for multiple testing applied to the p-values, to determine which native fish species changed significantly over time [50].

## Linking FCN Attributes to Fish Distribution Data

Multiple FCNs can occur in each HUC-12 (Fig 2). To link FCN attributes to the HUC-12 fish distribution data, we identified the longest historical and current FCN occurring in each HUC-12 and used their hydrography, physical habitat, and condition attributes to characterize the potential historic and current habitat available to fish species in each HUC-12 (Fig 2D). For each HUC-12, we calculated the change in the length (km) and presence/absence of the physical habitat variables as percent change from current to historic with negative values indicating loss of the habitat from the historic FCN.

We used a length-weighted average to summarize the HWI condition measure and StreamCat hydrologic regulation local stream reach-scale variables for each HUC-12. For StreamCat dam density and dam storage measures for the upstream contributing area of each stream reach, we used the values corresponding to the NHDPlus v2 reaches with the largest drainage area in each HUC-12 (i.e., the HUC-12 outlet). ArcGIS Pro 3.5.2 (ESRI, Redlands, CA) was used to conduct spatial analyses and generate the maps used in this study.

## Relating Changes in FCN Characteristics to Fish Distribution Changes

To initially explore the relationship between HUC-12 TBI values, current FCN condition variables, and the change in FCN habitat variables from historic to current periods, we visualized the Spearman's rank correlation coefficient (Spearman's *p*) (S3 Table). The strength and direction of the correlation relationships supported our hypotheses around the importance of connectivity, physical habitat diversity, and current habitat condition (S1 Table). We used random forest (RF) models to further understand how the suite of condition and physical attributes (i.e., FCN attributes) influenced TBI and how the relationships varied by ecoregion. Given the approach we used to link current and historic FCN attributes to HUC-12 watersheds (i.e., because FCNs can overlap with multiple HUC-12s) and due to inherent similarities in fish communities in adjacent HUC-12s watersheds, we were concerned about the impact of spatial autocorrelation on the reliability of the RF models. We used the *spatialRF* package [52,53] in R to generate Moran's I plots of TBI and the predictor variables at different distance thresholds and confirmed positive and significant spatial autocorrelation existed.

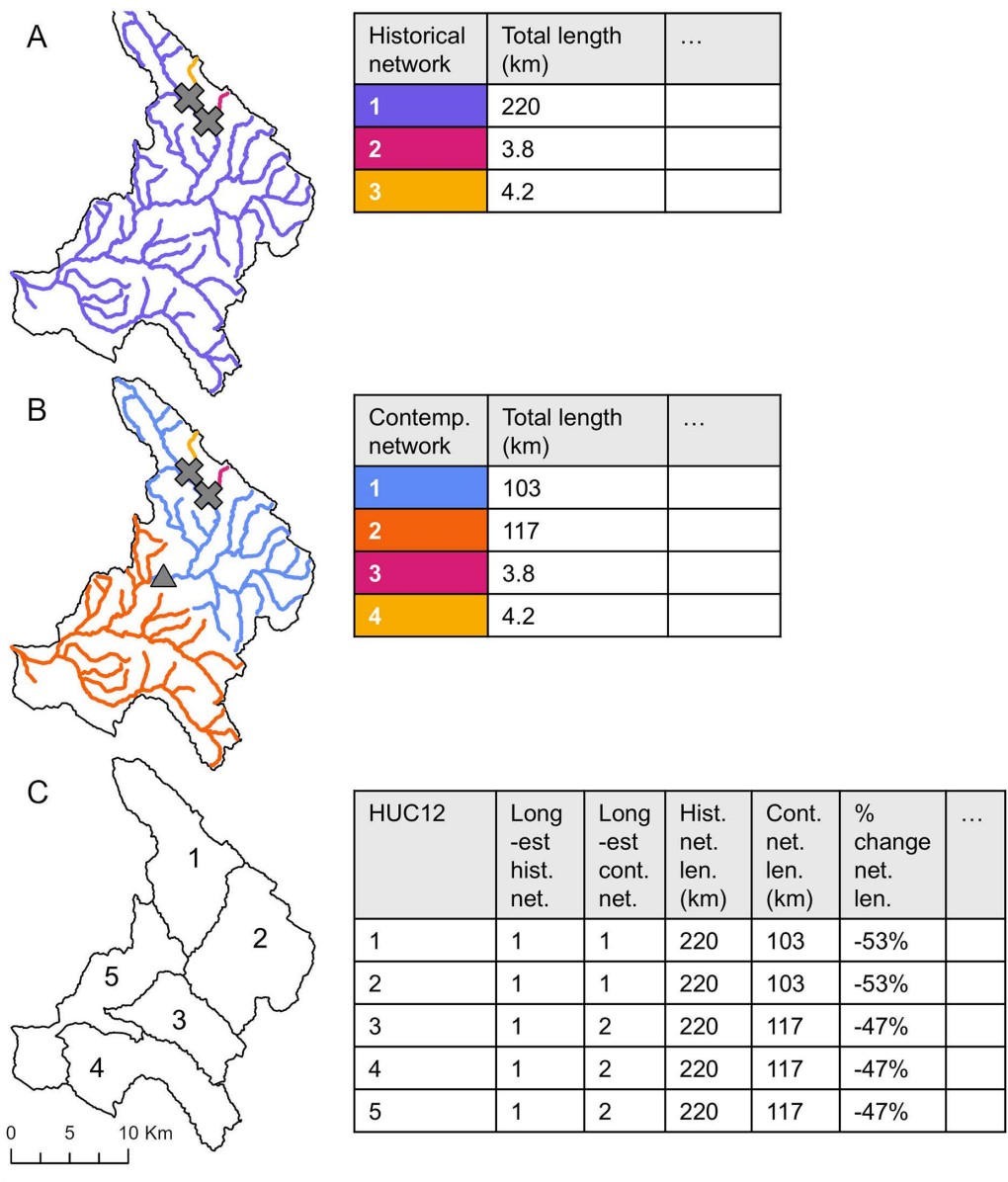

**Fig 2. Example networks in the Arroyo Grande watershed. A)** Historical functionally connected networks (FCN) (n = 3, fragmented by two waterfalls) and example attributes. **B)** Current FCNs (n = 4, fragmented by two waterfalls and one dam) and example attributes. **C)** HUC-12 subwatershed boundaries (n = 5) with example attributes, with the network attributes assigned based on the longest historical or current FCN intersecting each HUC-12.

To minimize the spatial autocorrelation of the RF model residuals and understand the influence of space on the response variable and its relationship with the FCN attributes, we used the *spatialRF* package to predict HUC-12 TBI values with the current habitat condition variables and the change in FCN habitat variables using spatial regression models with RF. We used the centroids of the HUC-12s as the coordinates in a Euclidean distance matrix and explored spatial autocorrelation for a series of distance thresholds that varied by ecoregion and were based on spatial data exploration of the response and predictor variables. For each ecoregion, we used the *auto-cor* and *auto-var* functions in the *spatialRF*

package with a variance inflation factor (VIF) threshold of 5 and a correlation threshold of 0.75 to remove highly correlated and multicollinear predictor variables (S4 Table). We used *the_feature_engineer* function in the *spatialRF* package to explore variable interactions but found they did not improve model performance as measured by the RF out-of-the-bag (OOB) $R^2$ values and were ecologically unintuitive.

After removing highly correlated variables (> 0.75), we ran a non-spatial RF and used the Moran's I index to evaluate the spatial autocorrelation of the RF residuals for the different distance thresholds [52,53] (S1-S4 Figs). After finding significant positive spatial autocorrelation in the RF residuals for all ecoregions except the Deserts-Lahontan, we ran spatial RFs using the *rf_spatial* function in the *spatialRF* package to successfully minimize the spatial autocorrelation of the model residuals. We also explored tuning the RF model to improve performance but found the default parameters were sufficient. For each ecoregion, the spatial RF models were run 30 times to better understand the distribution of predictor variable importance scores. Lastly, we used the *plot_response_curves* function to plot partial dependence curves for the most important RF predictors in each ecoregion. The curves were built by centering the other predictors to their 0.5 quantiles across all 30 RF models to elucidate how each predictor variable relates to the TBI values.

## Results

### FCN Delineation

We identified 228,883 km of streams, 838 natural barriers, and 1,847 anthropogenic barriers in the study area (Table 1). The number of FCNs within the study area increased by 59% from the historical to current periods, from 2,711–4,321, while the average length of FCNs decreased by 37%, from 84 km in historical FCNs to 53 km in current FCNs. Percent change in FCN count and average length were greatest in the Sacramento-San Joaquin ecoregion (144% increase in number of FCNs, 59% decrease in average stream length) and smallest in the Deserts-Lahontan ecoregion (18% increase in count, 15% decrease in average length).

### Changes in fish distribution

The results of the temporal beta diversity index (TBI) analyses showed that all ecoregions experienced statistically significant species losses and no gains at the HUC-12 scale (Fig 3, Table 2). Among the ecoregions, the Northern California ecoregion had the lowest dissimilarity or change over time in the fish presence-absence data and Southern California had the highest (Table 2).

TBI values of 0 indicate the HUC-12 experienced no species change (loss or gain) from historic to current, while a value of 1 indicates the HUC-12 had no shared species from the historic to present time periods. Most HUC-12s that had fish data for both time periods experienced species loss (TBI > 0) (Table 2, Fig 3A, Fig 3B). Concentrations of significant fish species loss at the HUC-12 scale can be seen in the Klamath, Shasta, Trinity, and Garcia rivers in Northern California;

**Table 1. FCN characterization by freshwater ecoregion.**

| Ecoregion | Total stream length (km) | Historical FCN count | Current FCN count | Ave. Historical FCN length (km) | Ave. Current FCN length (km) |
|---|---|---|---|---|---|
| Northern California | 32,842 | 490 | 735 | 67.0 | 44.7 |
| Sacramento-San Joaquin | 83,579 | 597 | 1,459 | 140.0 | 57.2 |
| Deserts-Lahontan | 80,694 | 1,059 | 1,251 | 76.2 | 64.5 |
| Southern California | 31,768 | 565 | 876 | 56.2 | 36.2 |
| All ecoregions | 228,883 | 2,711 | 4,321 | 84.4 | 52.9 |

Table 1. Summary of the total length of all FCNs and the count and average length of networks in historical and current periods by ecoregion.

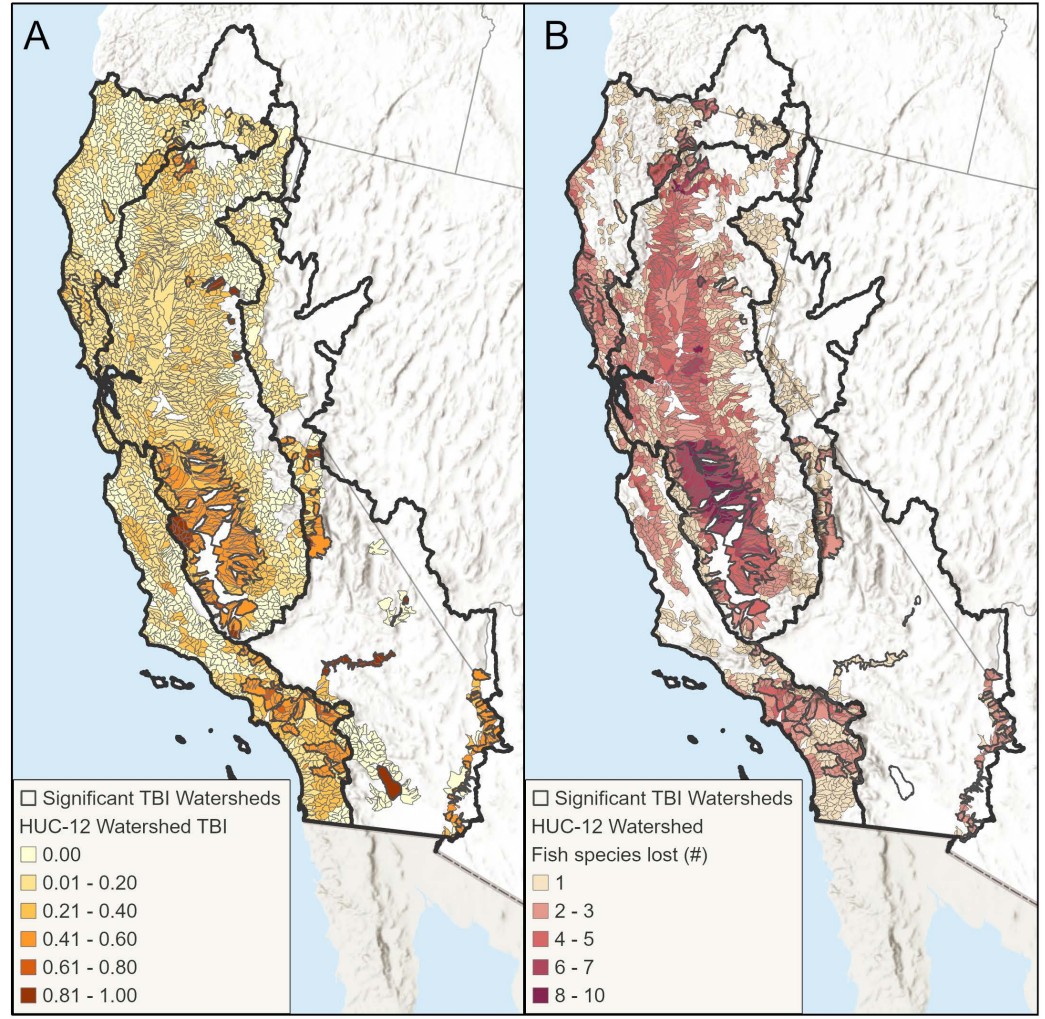

**Fig 3. Changes in Fish Distribution. A)** Map of TBI values for HUC-12 watersheds by California freshwater ecoregion. HUC-12s with a dark gray outline had significant TBI values (p≤ 0.05). **B)** HUC-12 watersheds by the number of fish species that were historically present but are now absent.

**Table 2. Summary of Temporal Beta-Diversity Indices (TBI) by California freshwater ecoregion (ER). The TBI analyses showed that all ecoregions experienced species loss over time.**

| Ecoregion | TBI loss | TBI dissimilarity | Mean TBI over all sites | # of HUC-12s with significant TBI values (p<.05) |
|---|---|---|---|---|
| Northern California | 0.065 | 0.065 | loss* | 66 |
| Sacramento-San Joaquin | 0.172 | 0.172 | loss* | 207 |
| Deserts-Lahontan | 0.184 | 0.184 | loss* | 55 |
| Southern California | 0.189 | 0.189 | loss* | 73 |

* significant at p <=.0001.

Table 2. TBI values for the four freshwater ecoregions.

the Sacramento and San Joaquin rivers in central California; the Los Angeles and Santa Ana rivers in Southern California; and the Mojave and Colorado rivers in the arid eastern part of the state (Fig 2).

All freshwater ecoregions experienced significant declines ($p \leq 0.05$) in individual fish species at the HUC-12 scale from the historical to the current time period (Table 3), and no species had significant increases over time. The Deserts-Lahontan ecoregion had the highest average (77%) and median (86%) percent decreases in the number of fish species that were historically present in HUC-12s, followed by Southern California with an average and median loss of approximately 60%, Sacramento-San Joaquin (ecoregion 2) with 50%, and 30% in Northern California (ecoregion 1). Of the 53 fish species that historically occurred in the Northern California ecoregion, species with significant losses (n = 14, or 26%) are predominately anadromous (n = 12). In the Sacramento-San Joaquin, 30 of the 60 fish species that occur in the region show declines (50%). Unlike Northern California, species with all range types (e.g., anadromous, narrow-ranging, wide-ranging) experienced declines in the Sacramento-San Joaquin. Anadromous species show significant declines in each ecoregion where they historically occurred, while numerous species including Central Coast Coho Salmon and tidewater goby have been extirpated from the Sacramento-San Joaquin ecoregion.

## Changes in FCN characteristics influence fish distribution over time

Based on average out-of-bag $R^2$ values across 30 runs, the spatial Random Forest models explained 79% of the variation in the temporal beta-diversity index in the Sacramento-San Joaquin ecoregion, 78% of the variation in Northern California, 72% in Southern California and 53% in the Deserts-Lahontan ecoregion (Fig 4, S5 Table). Models trained on 75% of the data explained between 8% to 60% of variation across the 30 training sets tested with 25% of the data (SI Table, S5 Table). While the most important predictors of TBI varied by ecoregion (Fig 4), incorporating spatial predictors improved the RF models for all ecoregions, except the Deserts-Lahontan ecoregion. The variable importance plots highlight similar themes in all the ecoregions and identify characteristics specific to each ecoregion as well (Fig 4). FCN fragmentation from historic to current periods, as measured by the percent change in stream and river length (km) within FCNs, was associated with higher TBI values (greater change in fish species composition at the HUC-12 scale due to loss of species) in the two coastal ecoregions (Northern California, Southern California) and in the arid ecoregion (Deserts-Lahontan). Current habitat conditions as measured by the HWI were important in all ecoregions, as were local and upstream measures of hydrologic alteration. The most important predictor variables in the heavily modified and agricultural Sacramento-San Joaquin ecoregion are local, reach-scale hydrologic regulation (presence and volume of reservoirs, agriculture and impervious land over and density of canals and ditches) followed by upstream dam density and dam storage. Fig 5 shows the spatial pattern of three variables (stream length, hydrologic alteration, and local hydrologic regulation) that were important predictors of fish species dissimilarity over time in multiple ecoregions.

To better understand how a single predictor variable relates to TBI when the other variables are centered, Figs 6-8 show partial dependence curves for the most important predictors for the Northern California, Sacramento-San Joaquin, and Southern California ecoregions. Given the relatively moderate RF performance for the Deserts-Lahontan ecoregion (53% variation explained) and the limited number of HUC-12s with fish species data (n = 358 or 22% of the ecoregion's HUC-12s), we did not conduct partial dependence curves for this region. To help interpret the partial dependence curves for each ecoregion, Fig 9 shows the spatial distribution of five key physical attributes (size, flow regime, gradient, temperature, and confinement) and habitat condition as measured by the HWI.

Higher TBI values and thus greater species dissimilarity (loss) over time in the Northern California ecoregion are associated with stream and river fragmentation from dams, poor habitat condition (HWI), and loss of access to medium sized rivers and reaches with stable baseflow (Fig 6). Higher TBI values are also associated with lower scores for the local hydrologic regulation metric from StreamCat. The importance of access to the ecoregion's topographic diversity is reflected by the loss of habitat with cool temperatures and steep gradients.

Table 3. Fish species decline by ecoregion.

| Ecoregion | Common Name | Range type | Count HUC-12s with historical occurrence | Count HUC-12s with current occurrence | Change in # of HUC-12s where species occur between historical and current periods | % Change in HUC-12 occurrence from historical period |
|---|---|---|---|---|---|---|
| Northern California | California Coast fall Chinook salmon | anadromous | 202 | 151 | −51 | −25.2 |
| Northern California | Central Coast coho salmon | anadromous | 125 | 79 | −46 | −36.8 |
| Northern California | Chum salmon | anadromous | 64 | 34 | −30 | −46.9 |
| Northern California | Coastal cutthroat trout | anadromous | 160 | 67 | −93 | −58.1 |
| Northern California | Eulachon | narrow | 55 | 16 | −39 | −70.9 |
| Northern California | Klamath Mountains Province summer steelhead | anadromous | 139 | 79 | −60 | −43.1 |
| Northern California | Klamath Mountains Province winter steelhead | anadromous | 243 | 212 | −31 | −12.8 |
| Northern California | Northern California coast summer steelhead | anadromous | 189 | 117 | −72 | −38.1 |
| Northern California | Pacific lamprey | anadromous | 538 | 506 | −32 | −6.00 |
| Northern California | River lamprey | anadromous | 102 | 75 | −27 | −26.5 |
| Northern California | Russian River tule perch | narrow | 37 | 18 | −19 | −51.3 |
| Northern California | Southern Oregon Northern California coast coho salmon | anadromous | 335 | 310 | −25 | −7.5 |
| Northern California | Upper Klamath-Trinity fall Chinook salmon | anadromous | 202 | 175 | −27 | −13.4 |
| Northern California | Upper Klamath-Trinity spring Chinook salmon | anadromous | 202 | 174 | −28 | −13.9 |
| Sacramento-San Joaquin | Bull Trout | narrow | 9 | 0 | −9 | −100 |
| Sacramento-San Joaquin | California golden trout | narrow | 20 | 10 | −10 | −50.0 |
| Sacramento-San Joaquin | Central California coast winter steelhead | anadromous | 74 | 66 | −8 | −10.8 |
| Sacramento-San Joaquin | Central Coast coho salmon | anadromous | 102 | 0 | −102 | −100 |
| Sacramento-San Joaquin | Central Valley fall Chinook salmon | anadromous | 188 | 125 | −63 | −33.5 |
| Sacramento-San Joaquin | Central Valley late fall Chinook salmon | anadromous | 185 | 119 | −66 | −35.7 |
| Sacramento-San Joaquin | Central Valley spring Chinook salmon | anadromous | 288 | 76 | −212 | −73.6 |
| Sacramento-San Joaquin | Central Valley steelhead | anadromous | 756 | 109 | −647 | −85.6 |
| Sacramento-San Joaquin | Central Valley winter Chinook salmon | anadromous | 81 | 33 | −48 | −59.2 |
| Sacramento-San Joaquin | Clear Lake Splittail | narrow | 10 | 0 | −10 | −100 |
| Sacramento-San Joaquin | Coastrange sculpin | wide | 41 | 0 | −41 | −100 |
| Sacramento-San Joaquin | Goose Lake redband trout | narrow | 26 | 19 | −7 | −26.9 |
| Sacramento-San Joaquin | Hardhead | wide | 592 | 495 | −97 | −16.4 |
| Sacramento-San Joaquin | Inland threespine stickleback | wide | 382 | 284 | −98 | −25.7 |
| Sacramento-San Joaquin | Kern brook lamprey | wide | 35 | 27 | −8 | −22.9 |
| Sacramento-San Joaquin | Kern River rainbow trout | narrow | 25 | 4 | −21 | −84 |
| Sacramento-San Joaquin | Northern (Pit) roach | narrow | 40 | 20 | −20 | −50 |
| Sacramento-San Joaquin | Pink salmon | anadromous | 19 | 3 | −16 | −84.2 |
| Sacramento-San Joaquin | Prickly sculpin | wide | 419 | 407 | −12 | −2.9 |

*(Continued)*

**Table 3.** (Continued)

| Ecoregion | Common Name | Range type | Count HUC-12s with historical occurrence | Count HUC-12s with current occurrence | Change in # of HUC-12s where species occur between historical and current periods | % Change in HUC-12 occurrence from historical period |
|---|---|---|---|---|---|---|
| Sacramento-San Joaquin | River lamprey | anadromous | 94 | 71 | −23 | −24.5 |
| Sacramento-San Joaquin | Sacramento blackfish | wide | 618 | 605 | −13 | −2.1 |
| Sacramento-San Joaquin | Sacramento hitch | wide | 615 | 331 | −284 | −46.2 |
| Sacramento-San Joaquin | Sacramento perch | wide | 400 | 0 | −400 | −100 |
| Sacramento-San Joaquin | Sacramento pikeminnow | wide | 1209 | 980 | −229 | −18.9 |
| Sacramento-San Joaquin | Sacramento speckled dace | wide | 280 | 104 | −176 | −62.9 |
| Sacramento-San Joaquin | Sacramento splittail | wide | 474 | 99 | −375 | −79.1 |
| Sacramento-San Joaquin | Sacramento tule perch | wide | 221 | 189 | −32 | −14.5 |
| Sacramento-San Joaquin | Thicktail Chub | wide | 464 | 0 | −464 | −100 |
| Sacramento-San Joaquin | Tidewater goby | wide | 9 | 0 | −9 | −100 |
| Sacramento-San Joaquin | White sturgeon | anadromous | 138 | 124 | −14 | −10.14 |
| Desert-Lahontan | Bonytail | wide | 40 | 0 | −40 | −100 |
| Desert-Lahontan | Colorado Pikeminnow | wide | 32 | 0 | −32 | −100 |
| Desert-Lahontan | Lahontan cutthroat trout | wide | 128 | 24 | −104 | −81.3 |
| Desert-Lahontan | Long Valley speckled dace | narrow | 8 | 1 | −7 | −87.5 |
| Desert-Lahontan | Mojave tui chub | narrow | 16 | 0 | −16 | −100 |
| Desert-Lahontan | Owens pupfish | narrow | 14 | 6 | −8 | −57.1 |
| Desert-Lahontan | Owens speckled dace | narrow | 44 | 15 | −29 | −65.9 |
| Desert-Lahontan | Owens sucker | wide | 58 | 49 | −9 | −15.5 |
| Desert-Lahontan | Owens tui chub | narrow | 22 | 3 | −19 | −86.4 |
| Southern California | Arroyo chub | wide | 169 | 41 | −128 | −75.7 |
| Southern California | Inland threespine stickleback | wide | 50 | 44 | −6 | −12 |
| Southern California | Monterey roach | wide | 141 | 30 | −111 | −78.7 |
| Southern California | Pacific lamprey | anadromous | 218 | 213 | −5 | −2.3 |
| Southern California | Sacramento perch | wide | 35 | 0 | −35 | −100 |
| Southern California | Sacramento tule perch | wide | 22 | 0 | −22 | −100 |
| Southern California | Santa Ana speckled dace | narrow | 56 | 22 | −34 | −60.7 |
| Southern California | Santa Ana sucker | wide | 30 | 14 | −16 | −53.3 |
| Southern California | South Central California coast steelhead | anadromous | 210 | 183 | −27 | −12.9 |
| Southern California | Southern California steelhead | anadromous | 315 | 63 | −252 | −80 |
| Southern California | Thicktail Chub | wide | 16 | 0 | −16 | −100 |
| Southern California | Tidewater goby | wide | 68 | 51 | −17 | −25 |
| Southern California | Unarmored threespine stickleback | narrow | 29 | 12 | −17 | −58.6 |

**Table 3.** Fish species that experienced significant declines (p < 0.05) from historic to current by California freshwater ecoregion (ER). Numeric values indicate the number of HUC-12s with the species for the two time periods. For example, Eulachon is no longer present in approximately 71% of HUC-12s in the Northern California ecoregion that historically had the species.

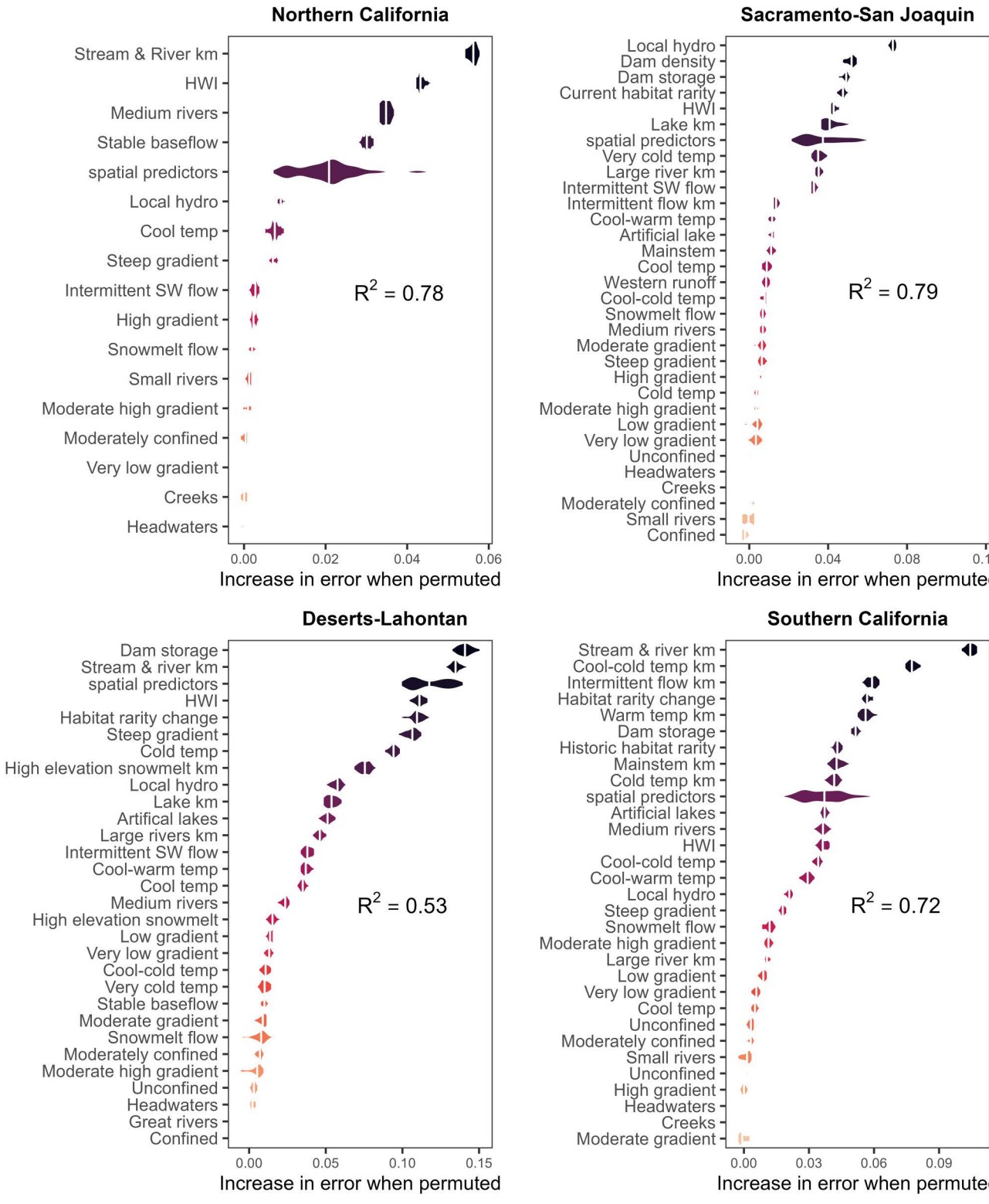

**Fig 4. Spatial Random Forest variable importance plots.** Variable importance plots for spatial RF by California freshwater ecoregion. Variables with "km" indicate percent change in the attribute length from current to historic while variables without "km" indicate change in presence/absence from current to historic.

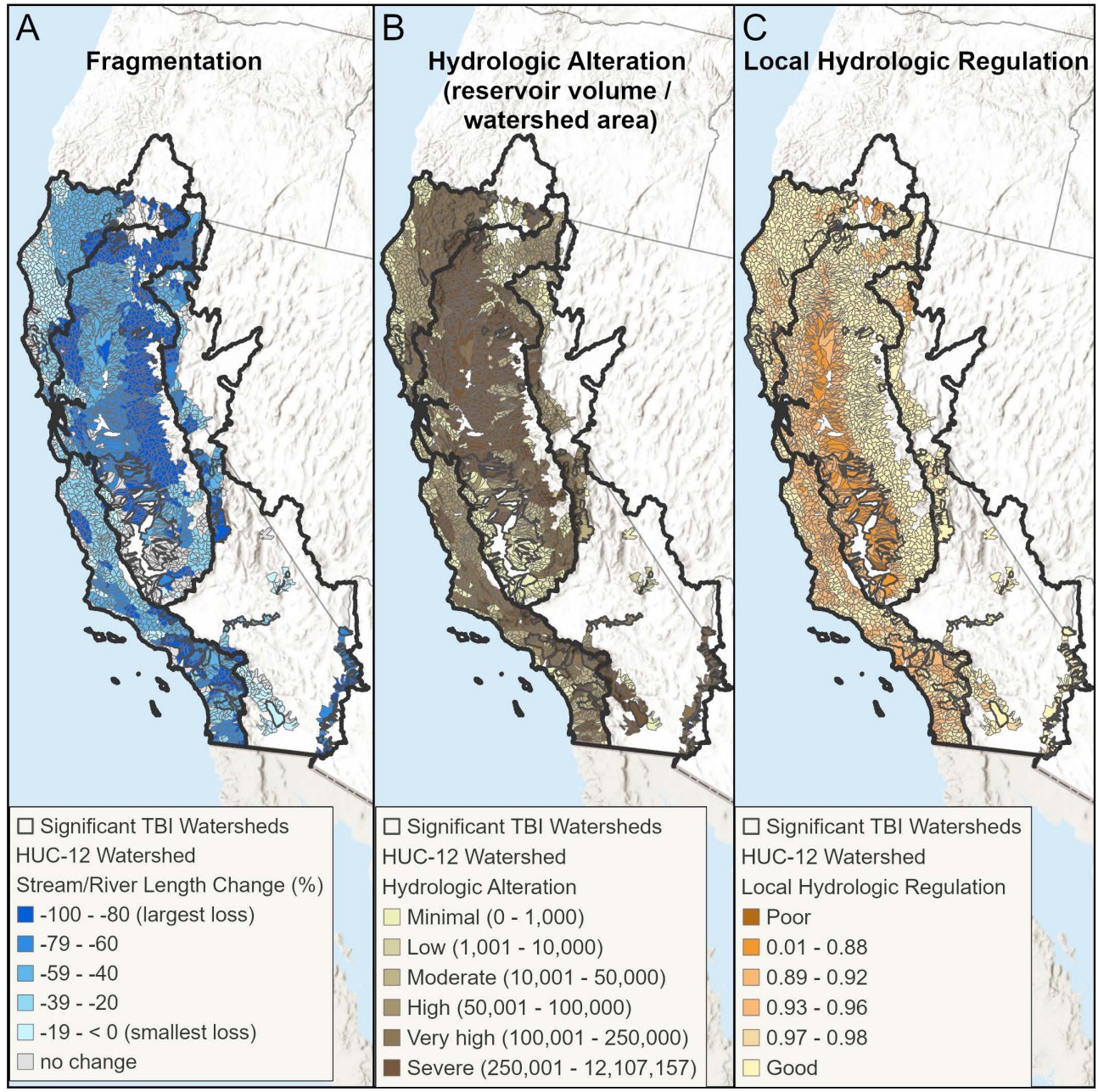

**Fig 5. Maps of important predictors of watershed TBI values in all ecoregions.** Maps of important predictors of watershed TBI values in all ecoregions with A) fragmentation as measured by the percent change in stream and river length (km) from historic to current; B) hydrologic alteration measured by the potential reservoir volume per upstream watershed area; and C) local hydrologic regulation from the EPA StreamCat dataset. HUC-12 watersheds with significant TBI values (p ≤ .05) are dissolved and shown in a dark gray border. World Hillshade is used in the figure's background (Sources: Esri, Airbus DS, USGS, NGA, NASA, CGIAR, N Robinson, NCEAS, NLS, OS, NMA, Geodatastyrelsen, Rijkswaterstaat, GSA, Geoland, FEMA, Intermap, and the GIS User Community).

Local and watershed measures of hydrologic alteration were the most important predictors of high TBI values in the Sacramento-San Joaquin ecoregion (Fig 7). Lower hydrologic regulation (i.e., more hydrological alteration) at the local scale was associated with greater species loss over time. As this metric includes agricultural land percentages, it also

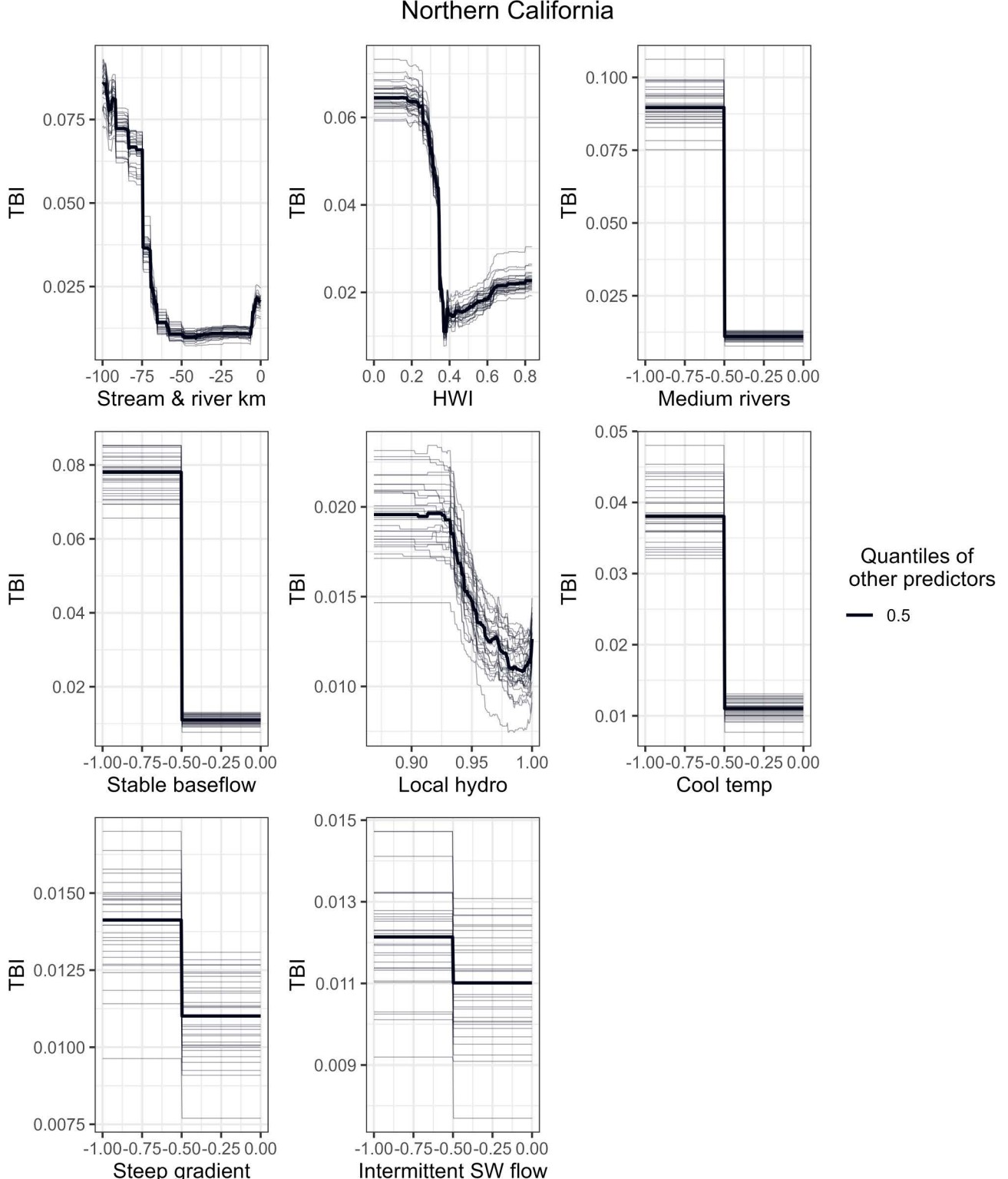

**Fig 6. Partial Dependence Curves for Northern California ecoregion.** Partial dependence curves between TBI and the 8 most important predictor variables across all 30 RF models for the Northern California ecoregion. The plots accentuate a variable's response curve by holding the other predictors to their 0.5 quantiles.

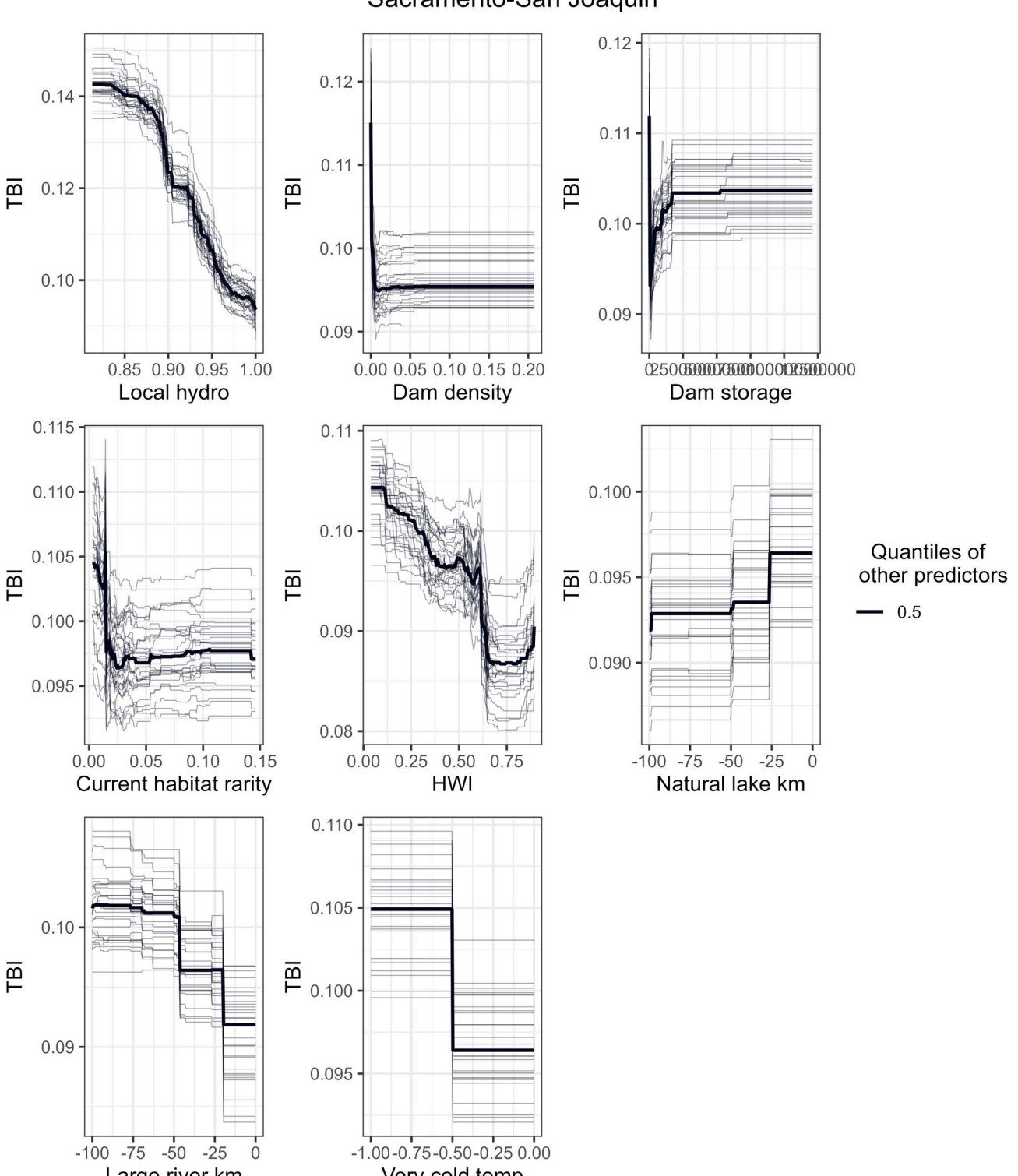

**Fig 7. Partial Dependence Curves for Sacramento-San Joaquin ecoregion.** Partial dependence curves between TBI and the top 8 most important predictor variables across all 30 RF models for the Sacramento-San Joaquin ecoregion. The plots accentuate the curve by holding the other predictors to their 0.5 quantiles.

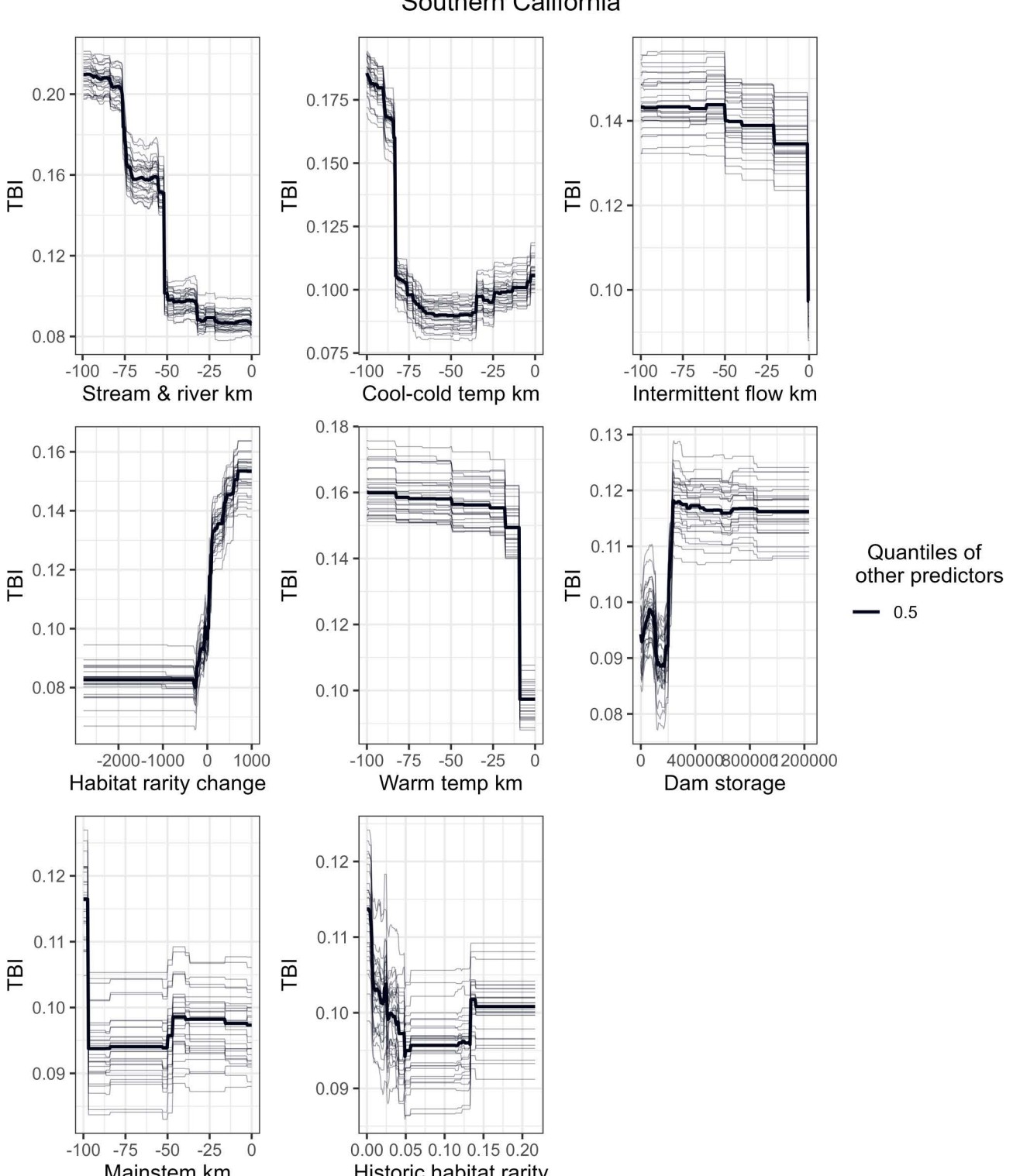

**Fig 8. Partial Dependence Curves for Southern California.** Partial dependence curves between TBI and the top 8 most important predictor variables across all 30 RF models for the Southern California ecoregion. The plots accentuate the curve by holding the other predictors to their 0.5 quantiles.

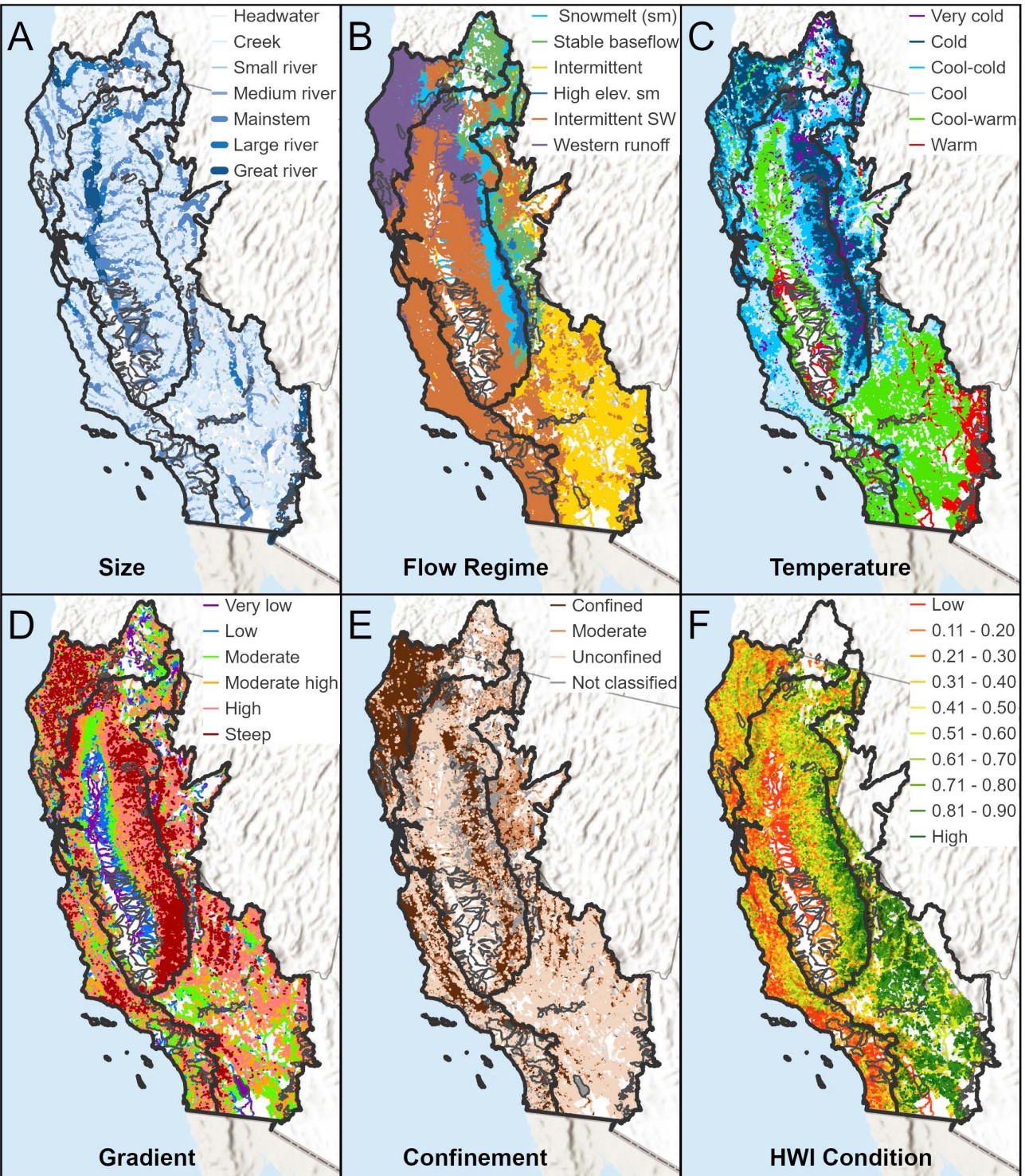

**Fig 9. Physical attribute maps.** To show how physical factors and habitat condition vary in each ecoregion, reach-scale values are mapped for the physical attributes of **A)** Drainage Area, **B)** Flow Regime, **C)** Temperature, **D)** Gradient, and **E)** Valley Confinement. In **F)**, continuous values of habitat condition from the HWI are grouped in equal-interval classes. HUC-12 watersheds with significant TBI values (p ≤ .05) are dissolved and shown in a dark gray border. World Hillshade is used in the figure's background (Sources: Esri, Airbus DS, USGS, NGA, NASA, CGIAR, N Robinson, NCEAS, NLS, OS, NMA, Geodatastyrelsen, Rijkswaterstaat, GSA, Geoland, FEMA, Intermap, and the GIS User Community).

reflects the predominance of agricultural activities in this region which is also captured in the response curve for the HWI condition index where poor condition is associated with higher TBI values. A map of the HWI (Fig 9) shows how the Central Valley scores poorly for condition relative to the rest of the state.

Fragmentation of historic FCNs was by far the most important predictor of high species dissimilarity over time in the Southern California ecoregion (Figs 8-9). Loss of FCN connectivity to reaches with cool-cold temperatures and intermittent flow regime were the second and third most important predictors.

## Discussion

This study evaluates factors associated with loss of fish species between historical and current periods within functionally connected networks to better understand factors important for resilience into the future. We found that changes in size and availability of certain habitat characteristics of historic FCNs has limited the ability of fish to persist throughout California's freshwater ecoregions. Over the past 50 years, each of the state's freshwater ecoregions have many HUC-12s where fish species that were historically found are no longer present. In the state's Sacramento-San Joaquin ecoregion, half (50%) of the fishes have experienced declines in the number of watersheds in which they historically occurred. In the Southern California ecoregion the loss is 48% and losses total 27% and 26% in the Deserts-Lahontan and Northern California ecoregions, respectively. California's FCNs have been heavily modified to support vast agricultural/urban/industrial economies and a growing human population, and unsurprisingly, our results show that FCN connectivity, hydrologic alteration, and habitat condition are the dominant factors in driving fish persistence.

The availability of historic and current native fish species data in California and the Temporal Beta Index (TBI) method were critical to quantify the change in fish species composition and individual species change over time. Previous studies have established the relationship between aquatic macroinvertebrate or plant biodiversity and habitat diversity based on stream-reach scale data [24–26] or used patterns of historical and current fish distribution in California to inform the design of protected areas [49], but this study is unique in quantitatively linking declines in species distribution to changes in habitat diversity and condition, and connectivity. The results provide support that current conservation efforts aimed at reconnecting FCNs by removing barriers (e.g., dams) and reconnecting aquatic habitat is critical to enhancing freshwater species persistence (in this case fishes) by increasing FCN length and habitat diversity.

This study, though lacking mechanistic details, points to key drivers of fish persistence [54]. For example, in both Southern and Northern California ecoregions, change in stream length in FCNs between historical and current periods ranks highest in explaining declines in fish community composition at the HUC-12 scale. Fish declines in those regions are dominated by anadromous species (85% of species with significant losses at the HUC-12 scale, Table 3). However, other important factors explaining historical and current fish distribution changes differ between the two regions. Factors such as access to medium-sized rivers, reaches with stable baseflow, and local habitat condition rank high in the Northern California ecoregion, whereas access to reaches with cool-cold water temperatures and/or intermittent flow and change in habitat rarity are the predominant factors explaining fish community changes (TBI) in the Southern California ecoregion. For example, loss of cool-cold FCN connectivity occurred east of Oceanside in the ecoregion while fragmentation of reaches with intermittent flow regimes was centralized in the headwaters of the Santa Margarita River in the southeastern portion of the ecoregion (Fig 5). A closer look at the relationship between low TBI values and high habitat rarity change (Figure S-5) found that a small number of watersheds experienced a large loss of habitat rarity from historic to current, but these same HUC-12s had few fish species to begin with. Thus, the loss of a single species did not result in a large TBI relative to the species losses experienced in other HUC-12s in the Southern California ecoregion. HUC-12s near the city of Riverside lost historic FCN connectivity to warmwater streams and experienced significant TBI values. A cluster of HUC-12s with significant TBI values and fish species loss occurs in the Los Angeles metropolitan area and is characterized by very high flow alteration from dam storage relative to upstream drainage area (Fig 9). HUC-12s in the urban areas of Los Angeles, Long Beach, and Santa Ana also experienced loss of mainstem river habitat. Unsurprisingly, poor habitat

condition is also an important factor in high TBI values in parts of the ecoregion that have experienced high rates of urbanization over time (Fig 9).

In the Sacramento-San Joaquin ecoregion, hydrologic alteration was the most important factor in explaining fish loss, both at the local and upstream scales. While our findings of higher species losses with decreasing local hydrologic condition and increasing dam density and dam storage are consistent with our hypotheses that greater alteration of habitat is associated with increased species losses, we do also see counterintuitive results related to high species losses associated with the lowest dam density and dam storage values. The relationship between slightly higher TBI values and low upstream dam density in current FCNs is opposite of what might be expected, but a map of the values shows low dam density in the southern portion of the ecoregion where watersheds experienced the most species loss (Fig 3). This pattern appears related to hydrology in the ecoregion: species losses in the ecoregion are concentrated in the southern portion in the San Joaquin River valley, a region of extensive agricultural development, where stream reaches have been largely replaced by canals (Fig 7 A-7C). While the number of dams is lower in this part of the ecoregion, the potential dam storage relative to upstream drainage area is high (Fig 5B). The combination of poor watershed condition and high flow alteration likely impacted the species in the San Joaquin River. The spatial pattern of FCN attributes differing markedly between the northern and southern portions of the Sacramento-San Joaquin ecoregion is also evident in the other important predictor variables. For example, TBI values are slightly higher for HUC-12s where the rarity-weighted richness of habitat types for current FCNs is lower. Mapping this variable (S5_Figure) shows the highest and most homogenous values occur in a large cluster of the Sacramento River watersheds compared to the San Joaquin River watersheds which have the lowest current habitat rarity values (S5 Fig). Watersheds in the Sierra Nevada mountains have substantial variability in their current FCN habitat rarity values. The NHDPlus v2 stream reaches we used for representing hydrography poorly represent historical connectivity between streams and upstream areas in this ecoregion. Spatial data of historical stream hydrography and habitats are lacking but could enable more robust accounting of change in habitat availability between current and historical periods.

Similar limitations with the dataset used for representing habitats and habitat diversity are present in the Deserts-Lahontan ecoregion. There, the number of HUC-12s with fish species data was limited, and likely important drivers of fish species declines were not represented with our FCN attributes, such as groundwater and water availability. The substantial decline of groundwater in California and globally has received increased attention [55] but regionally consistent spatial data on groundwater withdrawals is limited. In addition, many of the fish species in this ecoregion have extremely small habitat ranges (e.g., pupfish) and the use of FCNs may be more relevant for anadromous and wide-ranging species. Furthermore, connected FCNs were delineated based on physical connectivity without considering if there is actually flow to connect those habitats or the probability of water availability. For example, many of the desert networks are large and 'connected' habitats that would not have had a historical hydrological connection. Nonetheless, the RF model for this ecoregion did show that loss of FCN connectivity was associated with significantly higher species composition change at the HUC-12 scale, a pattern that is most evident in the HUC-12s of the Colorado River where substantial flow alteration exists (Figs 4, 5).

Freshwater systems are strongly influenced and shaped by spatial processes and patterns. From the flow of water to the influence of human activities (dams, agricultural and urban runoff, etc.), watersheds near each other typically have more similar hydrologic, chemical, and ecological characteristics than watersheds further apart. As Figs 3A and B show, watersheds with significant TBI values and species loss from historic to current periods tend to be clustered. By representing the influence of spatial processes and patterns with spatial predictors in a RF framework, we were able to explore the impact of dam construction, land use change, and flow modification on fish community change in California's HUC-12 watersheds.

While the TBI results definitively show California's HUC-12s experienced loss of individual fish species over time, the use of presence-absence data likely underestimates the scope of loss. Abundance data would provide important

information on population size and trends, allowing assessment of the future viability of a species. Furthermore, it is important to note our key assumption that the presence-absence of native fish species in HUC-12s is related to the longest historic and current FCN. We selected the longest historic and current FCN that intersected a HUC-12 to represent the potential full suite of habitat options available to a fish in an FCN for the two time periods but this is likely less relevant for fish with limited dispersal ability. Future efforts to refine fish distribution information at finer scales could be used to link fish to FCNs.

This study underscores the importance of stream connectivity and habitat diversity in sustaining freshwater fish communities. Our findings support conservation strategies that enhance freshwater resilience by restoring connectivity – such as dam removal – and targeting habitat types linked with the greatest species losses. In Northern California, medium rivers and stable baseflows, and in Southern California cool-cold temperature and intermittent-river flows offer high potential for fish species persistence and resilience. California's 2024 removal of four dams on the Klamath River, reconnecting ~650 kilometers of stream habitat, exemplifies this approach; fall-run Chinook salmon were observed upstream of the JC Boyle Dam site (the most upstream dam removed) for the first time since 1912. With nearly 2,000 dams fragmenting California's rivers, this study affirms the urgency of reconnecting freshwater networks to maintain biodiversity and ecosystem function [56]. Moving forward, conservation efforts should focus on preserving large, contiguous freshwater systems and mitigating the effects of habitat fragmentation through targeted restoration actions. Incorporating connectivity metrics and habitat assessments into conservation planning can strengthen ecological resilience, ensuring that riverine systems continue to support diverse and adaptable fish populations. As climate change accelerates, proactive management strategies will be essential in safeguarding California's freshwater biodiversity and preventing further declines in species persistence.

## Supporting information

**S1 Fig. Distance thresholds – Northern California.** Moran's I plots of the residuals for distance thresholds used in the non-spatial RF (first panel) and the spatial RF (second panel) for the Northern California ecoregion. P-values greater than 0.05 are considered non-significant for spatial autocorrelation and are found in the spatial RF.
(TIF)

**S2 Fig. Distance thresholds – Sacramento-San Joaquin.** Moran's I plots of the residuals for distance thresholds used in the non-spatial RF (first panel) and the spatial RF (second panel) for the Sacramento-San Joaquin ecoregion. P-values greater than 0.05 are considered non-significant for spatial autocorrelation and are found in the spatial RF.
(TIF)

**S3 Fig. Distance thresholds – Deserts-Lahontan.** Moran's I plots of the residuals for distance thresholds used in the non-spatial RF (first panel) and the spatial RF (second panel) for the Deserts-Lahontan ecoregion. P-values greater than 0.05 are considered non-significant for spatial autocorrelation and are found in both the non-spatial and spatial RF.
(TIF)

**S4 Fig. Distance thresholds – Southern California.** Moran's I plots of the residuals for distance thresholds used in the non-spatial RF (first panel) and the spatial RF (second panel) for the Southern California ecoregion. P-values greater than 0.05 are considered non-significant for spatial autocorrelation and are found in the spatial RF.
(TIF)

**S5 Fig. Rarity-weight habitat richness.** The rarity-weighted habitat richness of historic (A) and current networks (B) for HUC-12 watersheds is shown by z-score classes (Table S-5) relative to California freshwater ecoregions. Z-scores were used to facilitate comparison of the richness values for the two time periods but were not a perfect solution as discussed in the main text. Panel C shows the change in z-scores from current to historic networks where negative values indicate lower rarity-weighted richness in the current network. World Hillshade is used in the figure's background (Sources: Esri,

Airbus DS, USGS, NGA, NASA, CGIAR, N Robinson, NCEAS, NLS, OS, NMA, Geodatastyrelsen, Rijkswaterstaat, GSA, Geoland, FEMA, Intermap, and the GIS User Community).
(TIF)

**S1 Table. Stream reach criteria.** Categories used to classify stream reaches based on gradient, valley confinement, and flow regime.
(DOCX)

**S2 Table. Freshwater fish species.** List of the 129 native freshwater fish species used to conduct this analysis.
(DOCX)

**S3 Table. Correlation coefficients.** Spearman correlation coefficients between HUC-12 TBI values and all HUC-12 physical habitat and condition variables by California freshwater ecoregion. Correlation p-value $\leq 0.001 = ***, \leq 0.01 = **, \leq 0.05 = *$. Physical habitat variables with "km" are the percent change in length from current to historic while physical habitat variables without "km" are the change in presence/absence from current to historic. Condition variables are for current networks only. The first column for each ecoregion provides the correlation coefficient and p-value for the HUC-12s that had significant TBI values. The second column corresponds to all HUC-12 watersheds in each ecoregion.
(DOCX)

**S4 Table. Variables omitted.** Network variables by California freshwater ecoregion that were removed for the random forest analyses due to high correlation, multicollinearity, or zero variance. "Cor" indicates the variable exceeded the correlation threshold of 0.75. "VIF" identifies variables above the variance inflation threshold of 5. "ZV" refers to variables that had zero variance (i.e., values were identical for the ecoregion). Physical habitat variables with "km" are the percent change in length from current to historic while physical habitat variables without "km" are the change in presence/absence from current to historic. Condition variables are for current networks only.
(DOCX)

**S5 Table. Performance statistics for random forest models.** Performance statistics ($R^2$ and RMSE) for the full random forest models repeated 30 times for each California freshwater ecoregion. Out-of-bag (OOB) refers to the subset of the full dataset not used to build the individual decision trees of each RF model. The root mean squared error (RMSE) is the average difference between the actual values and those predicted by the RF model. The OOB values are the median and median absolute deviation (MAD) of the 30 model runs. Moran's I is a measure of spatial autocorrelation with the median value for all distance thresholds reported in the table below.
(DOCX)

**S6 Table. Z-scores.** Rarity-weighted habitat richness values of historic and current networks were translated to z-scores by freshwater ecoregion and assigned to classes using these ranges.
(DOCX)

## Author contributions

**Conceptualization:** Jeanette K Howard, Analie R. Barnett, Kurt A. Fesenmyer, Mark G. Anderson.

**Data curation:** Analie R. Barnett, Kurt A. Fesenmyer.

**Formal analysis:** Jeanette K Howard, Analie R. Barnett, Kurt A. Fesenmyer.

**Investigation:** Jeanette K Howard.

**Methodology:** Jeanette K Howard, Analie R. Barnett, Kurt A. Fesenmyer.

**Project administration:** Jeanette K Howard.

**Visualization:** Jeanette K Howard, Analie R. Barnett, Kurt A. Fesenmyer.

**Writing – original draft:** Jeanette K Howard, Analie R. Barnett, Kurt A. Fesenmyer.

**Writing – review & editing:** Jeanette K Howard, Analie R. Barnett, Kurt A. Fesenmyer, Mark G. Anderson.

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
