## [Decision Letter · Decision Letter 0]

13 Aug 2025

Dear Dr. Howard,

Thank you for submitting your manuscript to PLOS ONE. After careful consideration, we feel that it has merit but does not fully meet PLOS ONE’s publication criteria as it currently stands, requiring a minor revision. Therefore, we invite you to submit a revised version of the manuscript that addresses the points raised during the review process.

Provide more information about if/where the associated data can be obtained in the data availability statement. With this be available through the corresponding author, a database, other?

A bit more clarification is needed in the “Historical and Current Fish Distribution” methods to provide a full understanding of how fish species were targeted- of the 129 taxa, were they separated by different fish runs, as well (fall-run, spring-run, etc.), for relevant species?

We look forward to receiving your revised manuscript.

Kind regards,

Jason Magnuson, Ph.D.

Academic Editor

PLOS ONE

Journal Requirements:

3. We note that Figures 1,2,3,5, 9 and S5 in your submission contain [map/satellite] images which may be copyrighted. All PLOS content is published under the Creative Commons Attribution License (CC BY 4.0), which means that the manuscript, images, and Supporting Information files will be freely available online, and any third party is permitted to access, download, copy, distribute, and use these materials in any way, even commercially, with proper attribution. For these reasons, we cannot publish previously copyrighted maps or satellite images created using proprietary data, such as Google software (Google Maps, Street View, and Earth). For more information, see our copyright guidelines: http://journals.plos.org/plosone/s/licenses-and-copyright.

1. You may seek permission from the original copyright holder of Figure(s) [#] to publish the content specifically under the CC BY 4.0 license. 

4. We notice that your supplementary figures and tables are uploaded with the file type 'Figure'. Please amend the file type to 'Supporting Information'. Please ensure that each Supporting Information file has a legend listed in the manuscript after the references list.

5. We notice that your supplementary figures and tables are included in the manuscript file. Please remove them and upload them with the file type 'Supporting Information'. Please ensure that each Supporting Information file has a legend listed in the manuscript after the references list.

6. Please include captions for your Supporting Information files at the end of your manuscript, and update any in-text citations to match accordingly. Please see our Supporting Information guidelines for more information: http://journals.plos.org/plosone/s/supporting-information .

Reviewers' comments:

Reviewer's Responses to Questions

**Comments to the Author**

1. Is the manuscript technically sound, and do the data support the conclusions?

Reviewer #1: Yes

Reviewer #2: Yes

2. Has the statistical analysis been performed appropriately and rigorously?

Reviewer #1: Yes

Reviewer #2: Yes

3. Have the authors made all data underlying the findings in their manuscript fully available?

Reviewer #1: Yes

Reviewer #2: No

4. Is the manuscript presented in an intelligible fashion and written in standard English?

Reviewer #1: Yes

Reviewer #2: Yes

Reviewer #1: The stated goal of this study is to further the understanding of freshwater fish species persistence in a diverse array of watersheds, through the example of California and California-adjacent drainage basins, in the face of anthropogenic fragmentation and land management. To achieve this, a thorough study of the relationship between past (pre-1975) and current (post-1975) fish presence/absence data and local anthropogenic/environmental conditions was conducted through the use of Temporal Beta Index calculation and Random Forest variable importance assessment. My own expertise with respect to this study is mostly related to freshwater fish and their relationships to anthropogenic perturbations, and basic SDM-adjacent analyses, and I am less competent when it comes to the minutiae of spatial autocorrelation analyses on one hand, and interpretation of Spearman’s ranking coefficient on the other - but a bit of quick enlightenment on my part leads me to trust the analyses presented. This study’s major result is the likely importance of fragmentation and other human-lead disturbances in fish species loss in California, and it discusses the implications of such results - and the expected nuances. This study is notable : the extensive available data used in the study for environmental variables and fish presence guarantee its relevance at the local and international level, and its results are important : they rely on novel ideas and provide relevant and necessary information on anthropogenic impacts on freshwaters ecosystems and points to specific practices as likely causes for species loss on a wide geographical scale.

I consider the article’s form adequate, following the PLOS One author guidelines - although the heading level indication may require slight clarification - and only see minor modifications as necessary for publication in PLOS One. It seems scientifically sound as far as my knowledge allows, and rigorous in its testing and interpretations. Below are a few comments pertaining mainly to clarification in methodology and matters of discussion.

METHODS

Lines 152-154 : There is a slightly awkward use of the word “connectivity” in this study : you mention connectivity in a broader sense in your title and introduction as well as your discussion, which includes many concepts that would be classified under longitudinal connectivity and includes matters of fragmentation - an essential point of your work - but the “connectivity” layer of your variables includes a much narrower sense of distance to larger bodies of water. I would suggest either renaming this layer or provide a bit more explanation for what I would consider a strangely broad term for a very specific metric.

Lines 182-189 : With respect to the formation of FCNs - this question may stem from my own inexperience with the relevant datasets, please disregard if irrelevant - which were the criteria (if any) used to conclude functional disconnection by dams and waterfalls (height? Fish passage data? An aggregated metric?). Similarly, were all barriers kept for analysis in other processes (excluding the exceptions cited in the article, culverts, removed barriers etc.), or was height accounted for in some way? I am not suggesting any type of change in the building of the dataset, simply a bit more detail on something which was not immediately clear to me.

Lines 208-210 : My curiosity may stem from low knowledge of California watersheds, but were all recorded species in Californian rivers included? If not, how were they selected ? Did the inclusion of nonnative species pose a challenge? A bit of ichthyological context would be appreciated, even if it must be only a few phrases.

Lines 272-284 : A few words on your methods for obtaining variable importance data and partial dependence plots would be welcome - otherwise they are first mentioned in your results. You would benefit from it since I personally don’t seem to find that you overstep in terms of random forest interpretation, and providing clearer guidelines for work reproduction would prevent unfounded criticism.

RESULTS

Lines 309-311 : A very minor gripe, but the way you write about TBI results is a bit strange, almost suggesting that TBI calculation includes an appreciation of species gain and loss, also in Table2. A reader familiar with beta diversity calculation will be able to parse what you mean, but a philistine may misunderstand you (of course, since species loss is the only phenomenon recorded - more on that later - I may even suggest drastically simplifying this section and Table to clarify what was obtained and increase legibility).

LINES 433-487 : Only significant change advised. While I fully understand the purpose of this section of the results, they seem to bleed into the discussion below a bit too much - you mention historical and geographical explanations to environmental responses of fish communities - and would benefit from a drastic reduction, while these elements of discussion are transported below to strengthen your discussion even more.

Relatedly, DISCUSSION

Your discussion is much appreciated and nicely concludes your study. As mentioned above, a few elements of anthropogenic management and geographical context may be added. If it is possible and judged relevant by the authors, I would find interesting a short reflection or opening on the link between species range type and resilience, since anadromous species are clearly pointed as singularly affected in your results, and since you argue anadromous may be especially suited to this kind of approach. Again, this is to the discretion of the authors.

Reviewer #2: An interesting study that uses historic and present day fish distributions to investigate how patterns in species loss are related to changes in connectivity and land use or a large spatial area. Overall this is a nice study and paper so I have minor comments.

I could not evaluate whether the data are publicly available or not. The data availability statement does not state how the data will be made available once the manuscript is accepted. The authors could make the data available to reviewers via a private repository. A statement could also me made about where a reader can find the publicly available datasets the study makes use of.

Using the largest Functionally Connected River segment to represent habitat available/connectivity within each HUC-12 is a weakness that is acknowledged in the discussion.

The language in the discussion shifts from species loss and TBI to the use of fish species persistence. I think it is more appropriate to use species loss or decline in TBI to remain consistent with previous use and to better reflect how the introduction sets up the work.

Road culverts and water diversions were not included due to a lack of consistent data. There are likely a lot of road crossings in the study area with a side range in permeabilities for individual fish species. A good addition to the discussion would be a few sentences that acknowledge the limitation and state that road crossing can make up a large number of barriers, particularly on smaller streams where culverts, not bridges, are installed.

Line 51: I suggest using habitat heterogeneity instead of habitat options here.

Line 109: By options do the authors mean increased habitat heterogeneity? More specific language will better explain the emphasis.

Line 217: Does wide ranging mean >24,000 km2 and not anadromous? Please clarify in text.

Line 234: delete “best”

Line 254: Check consistent us of abbreviation “HUC-12” here and throughout

Line 286: Please include the threshold used to identify highly correlated variables

Table 2: Is the “mean TBI over all sites” column correct? There are no values, just an “*” in each cell which indicates significance. Based on the column title I expect to see values here.

Table 3 is very large. It may be more appropriate to put the table in an appendix and refer to main highlights in text as was done in the results.

Line 493-496: I think these sentences can be removed. They are redundant and not necessary to understand the findings of the study.

Lien 510: Are you referring to fish species data here? Please clarify.

Line 522: I don’t think there should be a reference to climate change here unless the intent is to state the variables identified as important are likely to change in response to climate change. If that is the case an explicit statement should be made.

Lines 585-610: These are two important, but long paragraphs, that state the contribution of the study to conservation in general. However, I think the should be edited and combined into one paragraph to concisely state the big picture conservation implications of the work. The majority of the paper is about linking changes in TBI to changes in connectivity and land use and the discussion should focus on these topics.

**Do you want your identity to be public for this peer review?** For information about this choice, including consent withdrawal, please see our Privacy Policy

Reviewer #1: No

Reviewer #2: No

---

## [Author Response · Author response to Decision Letter 1]

26 Nov 2025

Editor’s General Comments

Provide more information about if/where the associated data can be obtained in the data availability statement. With this be available through the corresponding author, a database, other?

JH: Spatial and tabular data for reproducing this analysis are available at doi.org/10.6084/m9.figshare.30096151.

A bit more clarification is needed in the “Historical and Current Fish Distribution” methods to provide a full understanding of how fish species were targeted- of the 129 taxa, were they separated by different fish runs, as well (fall-run, spring-run, etc.), for relevant species?

JH: We have clarified the methods for the fish species used in the study. We used all native fish species historically occurring in the state and provide a Table (S2 Table) to provide scientific and common names and range (anadromous, wide, narrow) in the table.

Editor’s Comments re Journal Requirements

JH: We have ensured that our manuscript and associated tables and figures meet PLOS ONE requirements.

JH: Spatial and tabular data for reproducing this analysis are available at doi.org/10.6084/m9.figshare.30096151.

3. We note that Figures 1,2,3,5, 9 and S5 in your submission contain [map/satellite] images which may be copyrighted. All PLOS content is published under the Creative Commons Attribution License (CC BY 4.0), which means that the manuscript, images, and Supporting Information files will be freely available online, and any third party is permitted to access, download, copy, distribute, and use these materials in any way, even commercially, with proper attribution. For these reasons, we cannot publish previously copyrighted maps or satellite images created using proprietary data, such as Google software (Google Maps, Street View, and Earth).

JH: We have revised the figures to meet PLOS ONE copyright restrictions by adding the following language: “World Hillshade is used in the figure’s background (Sources: Esri, Airbus DS, USGS, NGA, NASA, CGIAR, N Robinson, NCEAS, NLS, OS, NMA, Geodatastyrelsen, Rijkswaterstaat, GSA, Geoland, FEMA, Intermap, and the GIS User Community).”

4. We notice that your supplementary figures and tables are uploaded with the file type 'Figure'. Please amend the file type to 'Supporting Information'. Please ensure that each Supporting Information file has a legend listed in the manuscript after the references list.

JH: I have uploaded the supplementary materials under “Supporting Information.”

5. We notice that your supplementary figures and tables are included in the manuscript file. Please remove them and upload them with the file type 'Supporting Information'. Please ensure that each Supporting Information file has a legend listed in the manuscript after the references list.

JH: I have uploaded the supplementary materials under “Supporting Information” and added the legends to the supplementary materials in the manuscript after the references. Sorry for that mistake.

JH: Reviewers did not include recommendation to cite previously published works.

JH: I have reviewed the reference list and feel confident that it is complete and correct. I added three references to the manuscript since the initial submission to cite datasets we used to identify passage barriers in the functionally connected networks (Lines 177-186).

Reviewer #1: General comments:

The stated goal of this study is to further the understanding of freshwater fish species persistence in a diverse array of watersheds, through the example of California and California-adjacent drainage basins, in the face of anthropogenic fragmentation and land management. To achieve this, a thorough study of the relationship between past (pre-1975) and current (post-1975) fish presence/absence data and local anthropogenic/environmental conditions was conducted through the use of Temporal Beta Index calculation and Random Forest variable importance assessment. My own expertise with respect to this study is mostly related to freshwater fish and their relationships to anthropogenic perturbations, and basic SDM-adjacent analyses, and I am less competent when it comes to the minutiae of spatial autocorrelation analyses on one hand, and interpretation of Spearman’s ranking coefficient on the other - but a bit of quick enlightenment on my part leads me to trust the analyses presented. This study’s major result is the likely importance of fragmentation and other human-lead disturbances in fish species loss in California, and it discusses the implications of such results - and the expected nuances. This study is notable : the extensive available data used in the study for environmental variables and fish presence guarantee its relevance at the local and international level, and its results are important : they rely on novel ideas and provide relevant and necessary information on anthropogenic impacts on freshwaters ecosystems and points to specific practices as likely causes for species loss on a wide geographical scale.

JH: We appreciate the reviewer’s kind words and appreciate their noting the novel approach.

More specific comments:

METHODS

Lines 152-154 : There is a slightly awkward use of the word “connectivity” in this study : you mention connectivity in a broader sense in your title and introduction as well as your discussion, which includes many concepts that would be classified under longitudinal connectivity and includes matters of fragmentation - an essential point of your work - but the “connectivity” layer of your variables includes a much narrower sense of distance to larger bodies of water. I would suggest either renaming this layer or provide a bit more explanation for what I would consider a strangely broad term for a very specific metric.

JH: We agree with the reviewer that the use of connectivity in the context of adjacency to water bodies was awkward. We have removed the word “connectivity” when referring to networks containing larger water bodies. Our revision is as follows “Additionally, we created reach classes for indicating presence of three water body types: ocean/estuary, natural lakes greater than 2.5 km², and artificial lakes greater than 2.5 km².” We hope that clarifies what we intended.

Lines 182-189 : With respect to the formation of FCNs - this question may stem from my own inexperience with the relevant datasets, please disregard if irrelevant - which were the criteria (if any) used to conclude functional disconnection by dams and waterfalls (height? Fish passage data? An aggregated metric?). Similarly, were all barriers kept for analysis in other processes (excluding the exceptions cited in the article, culverts, removed barriers etc.), or was height accounted for in some way? I am not suggesting any type of change in the building of the dataset, simply a bit more detail on something which was not immediately clear to me.

JH: We agree with the reviewer that we needed to include a bit more information re passage barriers. In the Passage Barriers section, we added language to clarify and citations for datasets to define passage barriers as follows:

“For artificial barriers, we used dams identified as complete, partial, or unknown barriers to fish passage in a national dataset (43) and the three regional datasets noted above. We excluded dams that did not fall along the routed portion of the NHDPlus v2 streams (e.g., the dam was off-channel water storage) and dams identified as removed by a national database (44). For both natural and artificial barriers in the Northern California, Sacramento-San Joaquin, and Southern California ecoregions, we further excluded waterfalls or dams with observed anadromous fish distribution ((45,46) upstream of the barrier. We did not include road culverts or surface water diversions due to incomplete coverage of inventory data for these features in the study area and the impermanence of these features relative to dams and waterfalls.”

Lines 208-210 : My curiosity may stem from low knowledge of California watersheds, but were all recorded species in Californian rivers included? If not, how were they selected ? Did the inclusion of nonnative species pose a challenge? A bit of ichthyological context would be appreciated, even if it must be only a few phrases.

JH: This study included all native freshwater fish species that have historically and currently occurred in California. To clarify which species were considered, we provided a list in S2 Table. Nonnative species were excluded due to the lack of reliable data on their historical and current distributions.

Lines 272-284 : A few words on your methods for obtaining variable importance data and partial dependence plots would be welcome - otherwise they are first mentioned in your results. You would benefit from it since I personally don’t seem to find that you overstep in terms of random forest interpretation, and providing clearer guidelines for work reproduction would prevent unfounded criticism.

JH: To better explain the methods for obtaining variable important data, we added the following two sentences to the section entitled “Relating Changes in FCN Characteristics to Fish Distribution Changes:”

“Lastly, we used the plot_response_curves function to plot partial dependence curves for the most important RF predictors in each ecoregion. The curves were built by centering the other predictors to their 0.5 quantiles across all 30 RF models to elucidate how each predictor variable relates to the TBI values.”

RESULTS

Lines 309-311 : A very minor gripe, but the way you write about TBI results is a bit strange, almost suggesting that TBI calculation includes an appreciation of species gain and loss, also in Table2. A reader familiar with beta diversity calculation will be able to parse what you mean, but a philistine may misunderstand you (of course, since species loss is the only phenomenon recorded - more on that later - I may even suggest drastically simplifying this section and Table to clarify what was obtained and increase legibility).

JH: We modified the language in the text and Table 2 to clarify that we mean species loss in HUC-12s. We hope the changes we made are clarifying.

LINES 433-487 : Only significant change advised. While I fully understand the purpose of this section of the results, they seem to bleed into the discussion below a bit too much - you mention historical and geographical explanations to environmental responses of fish communities - and would benefit from a drastic reduction, while these elements of discussion are transported below to strengthen your discussion even more.

JH: We omitted the geographic areas and reduced the verbiage in the results. We moved some of the information previously under geographic ecoregions into the discussion which we hope strengthen the discussion.

Reviewer #2: General comments:

The language in the discussion shifts from species loss and TBI to the use of fish species persistence. I think it is more appropriate to use species loss or decline in TBI to remain consistent with previous use and to better reflect how the introduction sets up the work.

Road culverts and water diversions were not included due to a lack of consistent data. There are likely a lot of road crossings in the study area with a side range in permeabilities for individual fish species. A good addition to the discussion would be a few sentences that acknowledge the limitation and state that road crossing can make up a large number of barriers, particularly on smaller streams where culverts, not bridges, are installed.

JH: We thought hard and long about the reviewer’s comments re fish loss vs. fish persistence. In the introduction, we set up the stage to discuss biodiversity persistence – in this case fish. We decided to maintain our use of persistence in the discussion. We agree that road crossings and culverts can be impassable barriers to fish, however, those data are not available at the statewide scale.

Reviewer #2: Specific comments:

Line 254: Check consistent us of abbreviation “HUC-12” here and throughout

JH: We made changes to ensure we are consistent with the abbreviation HUC-12 instead of HUC12.

Line 286: Please include the threshold used to identify highly correlated variables

JH: We added the threshold used to identify highly correlated variables ((> 0.75).

Table 2: Is the “mean TBI over all sites” column correct? There are no values, just an “*” in each cell which indicates significance. Based on the column title I expect to see values here.

JH: We replaced “*” with the word “loss” to better reflect what we meant in that column.

Table 3 is very large. It may be more appropriate to put the table in an appendix and refer to main highlights in text as was done in the results.

JH: We appreciate that the table is long, but decided to keep it as is.

Line 493-496: I think these sentences can be removed. They are redundant and not necessary to understand the findings of the study.

JH: We removed the sentences.

Line 510: Are you referring to fish species data here? Please clarify.

JH: We added the word “data” here.

Line 522: I don’t think there should be a reference to climate change here unless the intent is to state the variables identified as important are likely to change in response to climate change. If that is the case an explicit statement should be made.

JH: We removed the reference to climate change.

Lines 585-610: These are two important, but long paragraphs, that state the contribution of the study to conservation in general. However, I think the should be edited and combined into one paragraph to concisely state the big picture conservation implications of the work. The majority of the paper is about linking changes in TBI to changes in connectivity and land use and the discussion should focus on these topics.

JH: We agree w

---

## [Editor Report · Decision Letter 1]

3 Dec 2025

From Fragmentation to Resilience: Connectivity and Habitat Diversity as Drivers of Fish Persistence in California Watersheds

PONE-D-25-34116R1

Dear Dr. Howard,

We’re pleased to inform you that your manuscript has been judged scientifically suitable for publication and will be formally accepted for publication once it meets all outstanding technical requirements.

Kind regards,

Jason Magnuson, Ph.D.

Academic Editor

PLOS ONE
---

## [Editor Report · Acceptance letter]

PONE-D-25-34116R1

PLOS One

Dear Dr. Howard,

I'm pleased to inform you that your manuscript has been deemed suitable for publication in PLOS One. Congratulations! Your manuscript is now being handed over to our production team.

Kind regards,

on behalf of

Dr. Jason Magnuson

Academic Editor

PLOS One